# Inhomogeneous Flow of Wormlike Micelles: Predictions of the Generalized BMP Model with Normal Stresses

**J. Paulo García-Sandoval** [1], **Fernando Bautista** [1], **Jorge E. Puig** [1] **and Octavio Manero** [2,*]

1   Departamentos de Física e Ingeniería Química, Universidad de Guadalajara, Blvd. M. García Barragán 1451, Guadalajara 44430, Jal., Mexico; paulo.garcia@cucei.udg.mx (J.P.G.-S.); ferbautistay@yahoo.com (F.B.); puig_jorge@hotmail.com (J.E.P.)

2   Instituto de Investigaciones en Materiales, Universidad Nacional Autónoma de México, Ciudad Universitaria, Ciudad de México 04510, Mexico

*   Correspondence: manero@unam.mx

**Abstract:** In this work, we examine the shear-banding flow in polymer-like micellar solutions with the generalized Bautista-Manero-Puig (BMP) model. The couplings between flow, structural parameters, and diffusion naturally arise in this model, derived from the extended irreversible thermodynamics (EIT) formalism. Full tensorial expressions derived from the constitutive equations of the model, in addition to the conservation equations, apply for the case of simple shear flow, in which gradients of the parameter representing the structure of the system and concentration vary in the velocity gradient direction. The model predicts shear-banding, concentration gradients, and jumps in the normal stresses across the interface in shear-banding flows.

**Keywords:** shear-banding flow; BMP model; normal stresses

## 1. Introduction

Beyond the local equilibrium hypothesis, the extended irreversible thermodynamics (EIT) provides a consistent methodology to derive constitutive equations for systems far from equilibrium. These equations, together with the conservation laws, predict flow-induced concentration changes produced by inhomogeneous stresses in complex fluids [1–4].

Flow produces changes in the internal structure of complex fluids and induces fluctuations in concentration and in the rheological properties. In some analyses of the rheology of these complex fluids, the stress constitutive equation couples with an evolution equation of a scalar representing the flow-induced modifications on the internal structure of the fluid (a variable such as the fluidity or micellar length, in the particular case of giant micellar solutions). Simultaneously, the stress coupling with the diffusion equation of the dispersed phase explains the phenomena arising from concentration changes, as suggested in reports on flow-induced concentration fluctuations and diffusive interfaces [5,6]. An additional coupling of diffusion and structural changes closes this scheme.

Rheological measurements in complex fluids, in particular those performed in wormlike micellar systems, demonstrate that a unique selected shear stress exists independently of flow history [7]. The steady-state flow curve has a well-defined, reproducible plateau. Coexistence of low and high viscosity bands has been observed by nuclear magnetic resonance (NMR) spectroscopy [8], small-angle neutron scattering (SANS) [9], and from flow birefringence [10], which reveals a highly oriented band coexisting with an isotropic one.

The Johnson-Segalman model (JS) predicts a history dependence of banded solutions after imposing several flow histories, i.e., the apparent flow curves and the stress plateau depend on

flow history. To find a unique stress selection, non-local gradient terms have been heuristically added to the JS constitutive equation [11,12], although diffusion in the stress arises naturally in kinetic theory, in particular in dumbbell models [13].

Although the non-local JS model has been useful to understand some features of the kinetics and stability of band formation, nevertheless there are two important setbacks of this model. The first one is that this model cannot describe the breaking and reformation processes of the micellar systems under flow to enable an understanding of the relation between shear-band formation and microstructural evolution. The second one refers to the inability of the model to describe the evolution of the stress and normal stress differences under step-strain experiments in shear flow and it gives wrong responses in extensional flow [14]. Furthermore, the non-local JS model may predict reversal in the band ordering in Couette flows [9] in contrast to experimental data.

Relevant alternative approaches, particularly that by Yuan and Jupp [15] using a 2-fluid J-S model, apparently give unique stress selection, even though interfacial terms were only present in the equation of motion for the concentration dynamics, and not in the constitutive viscoelastic equation. Vasquez et al. developed a two-species reptation-reaction network model [16] that captures the continuous breakage and reformation of long entangled chains that forms an entangled viscoelastic network, and the enhanced breaking that takes place during imposed shear deformation. The same group compared the homogeneous flow predictions of their model with steady and transient shear flow experiments performed in concentrated cetylpyridium chloride/sodium salicylate solutions [17]. In general, the predictions for nonlinear shear flow agree quite well with the rheological behavior for shear rates below the stress plateau, including the first normal stress difference, but it cannot predict a non-zero value of the second normal stress difference.

Concentration-coupling in the shear-banding transition of wormlike micellar systems has been a generic explanation of the slightly upward slope of the plateau stress observed with increasing shear rates [18–20]. This effect implies that a micellar concentration difference is established between the bands as the high-shear band grows to fill the gap. A model of concentration-coupled shear banding [21] was introduced by combining the diffusive (spatially non-local) J-S model [22] with a two-fluid approach to concentration fluctuations [23]. This model does not address the microscopic features of any particular viscoelastic system, but it should be regarded as a minimal model that combines an unstable constitutive curve, such as that of semi-dilute wormlike micelles, with spatially non-local terms in the viscoelastic constitutive equation and a simple approach to concentration coupling. In the two-fluid approach, the viscoelastic stress causes the micelles to diffuse up in stress gradients, and so it couples stress with concentration. If the viscoelastic stress then increases with concentration, positive feedback occurs, causing net diffusion of micelles up their own concentration gradient. This mechanism causes shear-enhanced concentration fluctuations and shear-induced de-mixing (concentration coupling) in systems that shear-band.

Normal stresses may be the reason for vorticity structuring that can emerge in complex fluids. A banding state may undergo a secondary linear stability due to the action of normal stresses across the interface between bands. Recent experiments of gradient-banded solutions of wormlike micelles show that they are unstable with respect to interfacial undulations with the wave vector in the vorticity direction [11,12]. The underlying mechanism includes normal stresses and shear-rate jumps across the interface.

In a previous work [24], we examined the non-homogeneous shear-banded flow of giant micelles with the BMP (Bautista-Manero-Puig) model. Results included the phase portraits around the flow curve and for the confined fluid, predictions were given for the velocity, stress, and fluidity fields as functions of both space and time. It was found that the same stress plateau and critical shear rates are approached independently of the initial conditions and shear history for a given applied shear rate or shear stress. The flow histories included forward and backward sweeps under strain-controlled and stress-controlled conditions.

In this work, the shear-banded flow predictions of the generalized BMP (Bautista-Manero-Puig) model [5], which contains the above-mentioned couplings, including normal stresses derived from EIT for complex fluids, are analyzed here. In reference [4], we present the formulation of the governing equations for flows that include normal stresses.

The derivation of the model's main concepts is given in the Appendix A and in [5]. The predictions of the phase portraits of the dynamics before approaching steady state and the variation of normal stresses in space and time of the resulting banded state are exposed. As in the previous paper, it is found that the same plateau is reached for various flow conditions; in this case, a stepping-up variation of shear rate is imposed. The diffusion equation allows for concentration gradients and gradients in the structural variable, and predicts stress gradient terms in the equation for the stress [13], which naturally arise in the constitutive equation, without the need to include them in an ad-hoc manner. Depleted regions where the concentration decreases are found near the moving boundary. It is shown that jumps in the normal stresses across the interface are also predicted.

## 2. Theoretical Description

The set of equations of the generalized BMP model are [5]:

$$\frac{\mathrm{d}\varphi}{\mathrm{d}t} = \frac{1}{\lambda}(\varphi_0 - \varphi) + k_0(1 + \vartheta(II_D))(\varphi_\infty - \varphi)\underline{\underline{\sigma}} : \underline{\underline{D}} + \varphi_0\beta_0'\nabla \cdot \bar{J} \tag{1}$$

$$\bar{J} + \tau_1\frac{\varphi_0}{\varphi}\overset{\nabla}{\bar{J}} = -\frac{\mathcal{D}\varphi_0}{\varphi}\nabla c - \frac{\beta_0}{\varphi}\nabla\phi + \frac{\beta_2\varphi_0}{\varphi}\nabla \cdot \underline{\underline{\sigma}} \tag{2}$$

$$\underline{\underline{\sigma}} + \frac{1}{G_0\phi}\overset{\nabla}{\underline{\underline{\sigma}}} = \frac{2}{\varphi}\underline{\underline{D}} + \frac{\psi_2}{\varphi}\underline{\underline{D}} \cdot \underline{\underline{D}} + \frac{\beta_2'\varphi_0}{\varphi}(\nabla\bar{J})^s \tag{3}$$

where $(\nabla\bar{J})^s$ stands for the symmetric part of $\nabla\bar{J}$ and the upper-convected derivatives of the diffusive concentration flux vector $\bar{J}$ and of the stress tensor $\underline{\underline{\sigma}}$ are defined, respectively, as:

$$\overset{\nabla}{\bar{J}} = \frac{\mathrm{d}\bar{J}}{\mathrm{d}t} - \underline{\underline{L}} \cdot \bar{J}, \tag{4}$$

$$\overset{\nabla}{\underline{\underline{\sigma}}} = \frac{\mathrm{d}\underline{\underline{\sigma}}}{\mathrm{d}t} - \left(\underline{\underline{L}} \cdot \underline{\underline{\sigma}} + \underline{\underline{\sigma}} \cdot \underline{\underline{L}}^T\right) \tag{5}$$

here $\mathrm{d}/\mathrm{d}t$ is the material-time derivative, $\underline{\underline{D}}$ is the symmetric part of the velocity gradient tensor $\underline{\underline{L}}$, and $II_D$ is its second invariant. $\varphi$ is the inverse of the shear viscosity ($\eta$) is the fluidity, $\varphi_0 \left(\equiv \eta_0^{-1}\right)$ is the fluidity at zero shear rate, $G_0$ is the plateau shear modulus, $\lambda$ is a structure relaxation time, and $k_0$ can be interpreted as a kinetic parameter for structure breaking. $\tau_1$ is a relaxation time for the mass flux, $\mathcal{D}$ is the Fickean diffusion coefficient, and $\psi_2$ is the second normal stress coefficient; $c$ is the local equilibrium concentration and $\vartheta$, $\beta_0$, $\beta_0'$, $\beta_2$, and $\beta_2'$ are phenomenological parameters. The structural variable $\sigma$ has been identified with the normalized fluidity $\phi = \varphi/\varphi_0$ [5].

Equations (1)–(3), together with the conservation equations, represent a closed set of time evolution equations for all the independent variables chosen to describe the behavior of complex fluids. Note the mutual coupling of these equations.

For simple-shear (where $x$ is the direction of the macroscopic flow velocity, $y$ is the direction of the velocity gradient, and $z$ is the vorticity direction), we assume small inertia and that the mass flux relaxation time is negligible compared to the stress relaxation time, i.e., $(G_0\varphi)^{-1} \gg \tau_1$ (which a plausible assumption for wormlike micelles). Equations (1)–(3) become:

$$\frac{\mathrm{d}\varphi}{\mathrm{d}t} = \frac{1}{\lambda}(\varphi_0 - \varphi) + k_0(1 + \vartheta(II_D))(\varphi_\infty - \varphi)\underline{\underline{\sigma}} : \underline{\underline{D}} + \varphi_0\beta_0'\left[\frac{\partial J_x}{\partial x} + \frac{\partial J_y}{\partial y}\right] \tag{6}$$

$$J_x = -\frac{\mathcal{D}\varphi}{\varphi_0}\frac{\partial c}{\partial x} - \frac{\beta_0}{\varphi}\frac{\partial \varphi}{\partial x} + \frac{\beta_2\varphi_0}{\varphi}\left[\frac{\partial \sigma_{xy}}{\partial y} + \frac{\partial \sigma_{xx}}{\partial x}\right] \tag{7}$$

$$J_y = -\frac{\mathcal{D}\varphi}{\varphi_0}\frac{\partial c}{\partial y} - \frac{\beta_0}{\varphi}\frac{\partial \varphi}{\partial y} + \frac{\beta_2\varphi_0}{\varphi}\frac{\partial \sigma_{xy}}{\partial y} \tag{8}$$

$$\sigma_{xy} + \frac{1}{G_0\varphi}\left[\frac{\partial \sigma_{xy}}{\partial t} - \dot{\gamma}\sigma_{yy}\right] = \frac{\dot{\gamma}}{\varphi} + \frac{\beta_2'\varphi_0}{\varphi}\left[\frac{\partial J_x}{\partial y} + \frac{\partial J_y}{\partial x}\right] \tag{9}$$

$$\sigma_{xx} + \frac{1}{G_0\varphi}\left[\frac{\partial \sigma_{xx}}{\partial t} - 2\dot{\gamma}\sigma_{yy}\right] = \frac{\beta_2'\varphi_0}{\varphi}\frac{\partial J_x}{\partial x} \tag{10}$$

$$\sigma_{yy} + \frac{1}{G_0\varphi}\left[\frac{\partial \sigma_{yy}}{\partial t}\right] = \psi_2\dot{\gamma}^2\frac{\varphi_0}{\varphi} + \frac{\beta_2'\varphi_0}{\varphi}\frac{\partial J_y}{\partial y} \tag{11}$$

$$\sigma_{zz} + \frac{1}{G_0\varphi}\left[\frac{\partial \sigma_{zz}}{\partial t}\right] = 0 \tag{12}$$

where $\dot{\gamma}$ is the shear rate. Equations (6)–(12) are the ones given particular attention in this work. To close the system of equations, the conservation of mass, concentration, and momentum are:

$$\rho\frac{\partial v_x}{\partial x} = 0 \tag{13}$$

$$\frac{\partial c}{\partial t} = -\left[\frac{\partial J_x}{\partial x} + \frac{\partial J_y}{\partial y}\right] \tag{14}$$

$$\rho\frac{\partial v_x}{\partial t} = \frac{\partial \sigma_{xy}}{\partial y} + \eta_s\frac{\partial^2 v_x}{\partial y^2} + \frac{\partial \sigma_{xx}}{\partial x}, \tag{15}$$

where $\eta_s$ is the solvent viscosity. In Equations (7), (8), and (15), the derivatives of the normal stresses involve terms of third order in the derivatives of the fluidity and concentration, which we neglect. In addition, a solution for Equations (13) and (15) can be obtained by taking the derivative of each term of Equation (15) with respect to $x$. This leads to:

$$\frac{\partial^2 \sigma_{xy}}{\partial x \partial y} + \frac{\partial^2 \sigma_{xx}}{\partial x^2} = 0 \tag{16}$$

Next, the normal stress differences are defined in the usual form:

$$N_1 = \sigma_{xx} - \sigma_{yy}, \ N_1 = \sigma_{yy} - \sigma_{zz} \tag{17}$$

From Equations (10)–(12) we obtain:

$$N_1 + \frac{1}{G_0\varphi}\left[\frac{\partial N_1}{\partial t} - 2\dot{\gamma}\sigma_{xy}\right] = \frac{\beta_2'\varphi_0}{\varphi}\left[\frac{\partial J_x}{\partial x} - \frac{\partial J_y}{\partial y}\right] \tag{18}$$

$$N_2 + \frac{1}{G_0\varphi}\left[\frac{\partial N_2}{\partial t}\right] = \psi_2\frac{\varphi_0}{\varphi}\dot{\gamma}^2 + \beta_2'\frac{\varphi_0}{\varphi}\frac{\partial J_y}{\partial y} \tag{19}$$

Until now, Equations (6), (9), (18), and (19) have been the more general expressions in two dimensions (the shear plane), including normal stresses. They preserve the tensorial character of the equations, upon the assumptions of small inertia and negligible mass flux relaxation time. Following the translational symmetry of the flow, we address the particular case where the derivatives in the direction of flow are negligible. In such case, we have:

$$\frac{\partial J_x}{\partial x} = \frac{\partial J_y}{\partial x} = 0 \tag{20}$$

$$\frac{\partial J_x}{\partial y} = \beta_2 \varphi_0 \frac{\partial}{\partial y}\left[\frac{1}{\varphi}\frac{\partial \sigma_{xy}}{\partial y}\right] \tag{21}$$

$$\frac{\partial J_y}{\partial y} = \frac{\partial}{\partial y}\left[\frac{1}{\phi}\left(-\beta_0 \frac{\partial \phi}{\partial y} + \beta_2 \phi_0 \frac{\partial \sigma_{yy}}{\partial y}\right)\right] + \frac{\partial}{\partial y}\left(\frac{-\mathcal{D}\phi_0}{\phi}\frac{\partial c}{\partial y}\right) \tag{22}$$

$$N_1 + \frac{1}{G_0\phi}\left[\frac{\partial N_1}{\partial t} - 2\dot{\gamma}\sigma_{xy}\right] = -\psi_2 \frac{\phi_0}{\phi}\dot{\gamma}^2 - \beta_2' \frac{\phi_0}{\phi}\left[\frac{\partial J_y}{\partial y}\right] \tag{23}$$

while (19) remains equal. Written in terms of the non-dimensional variables:

$$\boldsymbol{\phi} = \boldsymbol{\varphi}/\boldsymbol{\varphi_0}, \ \boldsymbol{\phi_\infty} = \boldsymbol{\varphi_\infty}/\boldsymbol{\varphi_0},$$

Equation (9) becomes:

$$\phi\sigma_{xy} + \tau_\sigma\left[\frac{\partial \sigma_{xy}}{\partial t}\right] = \eta_0\dot{\gamma} + \dot{\gamma}\tau_\sigma N_2 + \frac{\beta_2\beta_2'}{2}\frac{\partial}{\partial y}\left[\frac{1}{\phi}\frac{\partial \sigma_{xy}}{\partial y}\right] \tag{24}$$

where $\tau_\sigma$ is the stress relaxation time ($\tau_\sigma = (G_0\phi_0)^{-1}$) and $\eta_0$ is the zero shear-rate viscosity. In the limit of creeping flow $\phi \to 1$, Equation (24) reduces to:

$$\sigma_{xy} + \tau_\sigma\left[\frac{\partial \sigma_{xy}}{\partial t}\right] = \eta_0\,\dot{\gamma} + \frac{1}{2}\beta_2\beta_2'\frac{\partial^2 \sigma_{xy}}{\partial y^2} \tag{25}$$

Equation (25) is similar to the diffusion equation for the stress analyzed in the current literature to predict diffusion of interfaces [12,13,25]. In fact, following similar assumptions (simple shear, small inertia), Equation (25) is equal to that derived from the constitutive equation:

$$\underline{\underline{\sigma}} + \tau_\sigma\underline{\underline{\overset{\diamond}{\sigma}}} = 2\eta_0\underline{\underline{D}} + \tau_\sigma\mathcal{D}\nabla^2\underline{\underline{\sigma}} \tag{26}$$

where $\underline{\underline{\overset{\diamond}{\sigma}}}$ is the (Gordon-Schowalter) convected time derivative defined in the Johnson-Segalman model [26]. In creeping shear flows, Equation (26) reduces to Equation (25), providing the phenomenological coefficients $\beta_2\beta_2'$ to be identified with $\tau_\sigma\mathcal{D}$. This identification of the coefficients allows a physical interpretation and measurement of their magnitudes. Similarly, the coefficients $\beta_0\beta_0'$ may be identified with the structure diffusion coefficient $\mathcal{D}'$, and $\beta_0\beta_2'$ can be identified with $\rho\mathcal{D}\mathcal{D}'$. With these identifications, and considering that Equation (23) is decoupled, Equations (6), (9), (19), and (23) become:

$$\frac{\partial J_y}{\partial y} = \frac{\partial}{\partial y}\left[\frac{1}{\phi}\left(-\rho\mathcal{D}'\frac{\partial \phi}{\partial y} + \tau_\sigma\frac{\partial N_2}{\partial y}\right)\right] + \frac{\partial}{\partial y}\left(\frac{-\mathcal{D}}{\phi}\frac{\partial c}{\partial y}\right) \tag{27}$$

$$\frac{\partial \phi}{\partial t} = \frac{1}{\lambda}(1 - \phi) + k_0(1 + \vartheta\dot{\gamma})(\phi_\infty - \phi)\sigma_{xy}\dot{\gamma} + \frac{1}{\rho}\frac{\partial J_y}{\partial y} \tag{28}$$

$$\tau_\sigma\left[\frac{\partial \sigma_{xy}}{\partial t}\right] = \eta_0\dot{\gamma} + \dot{\gamma}\tau_\sigma N_2 + \frac{\mathcal{D}\tau_\sigma}{2}\frac{\partial}{\partial y}\left[\frac{1}{\phi}\frac{\partial \sigma_{xy}}{\partial y}\right] \tag{29}$$

$$\phi N_1 + \tau_\sigma\left[\frac{\partial N_1}{\partial t} - 2\dot{\gamma}\sigma_{xy}\right] = -\psi_2\dot{\gamma}^2 + \mathcal{D}\left[\frac{\partial J_y}{\partial y}\right] \tag{30}$$

$$\phi N_2 + \tau_\sigma\left[\frac{\partial N_2}{\partial t}\right] = \psi_2\dot{\gamma}^2 - \left[\frac{\partial J_y}{\partial y}\right] \tag{31}$$

Equations (28)–(31) are the main results of this section. It is noticeable that this formulation leads to structure-dependent variables, i.e., viscosity ($\eta_0/\phi$), stress relaxation time ($\tau_\sigma/\phi$), and diffusion coefficients ($\mathcal{D}/\phi$ and $\rho\mathcal{D}'/\phi$). The structure itself follows an evolution Equation (28). In the limit $\phi \to 1$ (constant structure) with no normal stresses, Equation (29) reduces to the simple-shear version of Equation (26). These equations contain seven constants. Five of them ($\lambda$, $k_0$, $\eta_0$, $\phi_\infty$, $\tau_\sigma$) can be

evaluated from independent rheological experiments [4]. Under heterogeneous (shear banding) flow, $\vartheta$ (the shear-banding intensity parameter) is related to the position of the stress plateau (set by the equal-areas criterion or equal minima in the dissipated energy) and $\mathcal{D}$ may be evaluated from data of interface diffusion. Dhont [27] has suggested an expression for the relaxed stress under simple shear (without normal stresses) similar to that of Equation (25), in which the term $(\mathcal{D}/\phi)$ is identified with the "curvature viscosity", which actually follows the same form of the shear-thinning viscosity observed in worm-like micellar solutions. According to the magnitudes shown in [27], usual values for the zero-shear rate viscosity are around 20 Pa·s. In the absence of normal stresses, the reaction-diffusion character of equation is preserved

$$\sigma_{xy} + \tau_\sigma \left[ \frac{\partial \sigma_{xy}}{\partial t} \right] = \eta_0 \dot{\gamma} + \frac{\mathcal{D}\tau_\sigma}{2} \frac{\partial}{\partial y} \left[ \frac{1}{\phi} \frac{\partial \sigma_{xy}}{\partial y} \right]$$

which reduces to the simple shear version of Equation (25) as $\phi \to 1$. It is worth mentioning that the inclusion of diffusion in the constitutive equations leads to a finite thickness of the interface between the bands, as shown in the results presented in the next section.

*2.1. Steady-State Solution*

Under steady state, the conservation Equations (14) and (15) lead to

$$\left[ \frac{\partial J_y}{\partial y} \right] = 0, \frac{\partial \sigma_{xy}}{\partial y} = 0$$

which means that both $J_y$ and $\sigma_{xy}$ are independent of the coordinates, and since there is no flux at the boundaries, then $J_y = 0$ for all $y$. Furthermore, Equation (8) leads to:

$$\rho \mathcal{D}' \left[ \frac{\partial \phi}{\partial y} \right] = -\mathcal{D} \left[ \frac{\partial c}{\partial y} \right] + \tau_\sigma \left[ \frac{\partial N_2}{\partial y} \right] \tag{32}$$

In the particular case when the mass and viscosity diffusion coefficients and the stress relaxation times are constant, Equation (32) can be integrated to give:

$$\phi = \phi_c - (\mathcal{D}/\rho \mathcal{D}')c + (\tau_\sigma/\rho \mathcal{D}')N_2$$

For a given constant $\phi_c$, the ratio of the diffusion coefficients is positive, and hence the fluidity increases with decreasing concentration. In addition, under steady state, Equations (28)–(31) become

$$0 = \frac{1}{\lambda}(1 - \phi) + k_0 (1 + \vartheta \dot{\gamma})(\phi_\infty - \phi)\sigma_{xy}\dot{\gamma} \tag{33}$$

$$\phi \sigma_{xy} = \eta_0 \dot{\gamma} + \dot{\gamma} \tau_\sigma N_2 \tag{34}$$

$$\phi N_1 = (2\tau_\sigma \sigma_{xy} - \psi_2 \dot{\gamma})\dot{\gamma} \tag{35}$$

$$\phi N_2 = \psi_2 \dot{\gamma}^2 \tag{36}$$

Substitution of Equations (36) and (34) in (33) leads to a fifth and third order equation for the shear rate and fluidity, respectively:

$$0 = (1 - \phi)\phi^2 + k_0 \lambda (1 + \vartheta \dot{\gamma})(\phi_\infty - \phi)\left( \eta_0 \phi + \tau_\sigma \psi_2 \dot{\gamma}^2 \right)\dot{\gamma}^2$$

In most cases, the order of magnitude of $\psi_2$ is negligible in comparison with $\eta_0 \phi$, and the previous equation reduces to a third and second order equation for the shear rate and fluidity, respectively, which has been previously analyzed [4].

*2.2. Numerical Method*

To analyze the transient behavior at the inception of flow predicted by the model, Equations (28)–(31) were numerically solved together with equations

$$\frac{\partial c}{\partial t} = -\frac{\partial J_y}{\partial y} \tag{37}$$

$$\rho \frac{\partial v_x}{\partial t} = \frac{\partial \sigma_{xy}}{\partial y} \tag{38}$$

which are, respectively, the simplified version, where the derivatives in the direction of flow ($x$ direction) as well as the solvent viscosity, $\eta_s$, are negligible. The moving plate is located at position $y = L$, while the fixed plate is at $y = 0$, with boundary conditions:

$$v_x(t, 0) = 0, \ v_x(t, L) = v_L(t) \tag{39}$$

$v_L(t) \geq 0$ is the upper plate velocity, which follows the following dynamics:

$$v_L(t) = v_{L,0} + (v_{L,ss} - v_{L,0})(1 - e^{-t/\tau_c}),$$

where $v_{L,0}$ is the initial velocity of the moving plate, $v_{L,ss}$ is the steady state velocity, and $\tau_c$ is a characteristic time. This time is linked to a controller, which regulates the velocity of the moving plate, which in general is of the same order of magnitude of the characteristic time of the system $\tau_\sigma$. In addition to the conservation law (Equation (38)) and mass conservation, we have to satisfy the following conditions:

$$\frac{\partial \sigma(t, 0)}{\partial y} = 0, \ \frac{\partial \sigma(t, L)}{\partial y} = \frac{1}{\rho} \frac{dv_L(t)}{dt} \tag{40}$$

$$J_y(t, 0) = 0, \ J_y(t, L) = 0 \tag{41}$$

As pointed out before [24], due to the conservation of momentum and the flow history $v_L(t)$, the spatial derivative of the stress on the upper plate cannot be zero. The boundary conditions in Equation (40) imply that the spatial derivative of the shear stress at the upper plate is zero only when steady state is reached. Thus, the boundary conditions at the upper plate are also history-dependent. In some cases, these boundary conditions give rise to a stable three-band state, similar to that found for Dirichlet boundary conditions, wherein the stress distribution in the gap depends on both the boundary condition and stress gradient [28,29].

Equations (28)–(31), (37), and (38), together with boundary conditions (39)–(41), are solved using the numerical method described in reference [24], where the ordinary differential equations resulting in discretization in space are integrated to obtain numerical solutions as a function of time.

## 3. Results

Figure 1 illustrates the variation of the stress, first normal stress difference ($N_1$), and the absolute value of the second normal stress difference ($N_2$) with shear rate, under shear-rate controlled flow starting from rest. The shear rate was increased in a stepwise mode allowing attainment of steady-state at each step. At the highest shear rate, $N_2$ approaches within a tenth of the value of $N_1$; this is the upper bound of $N_2$ as suggested by the scarce experimental data. $N_1$ shows a behavior similar to that of the stress-shear rate curve depicting, in fact, a shear-banding unstable region. In contrast, $N_2$ grows monotonically. For low shear rates, the slope of $N_1$ tends to a value of 2, although this range is not shown in Figure 1.

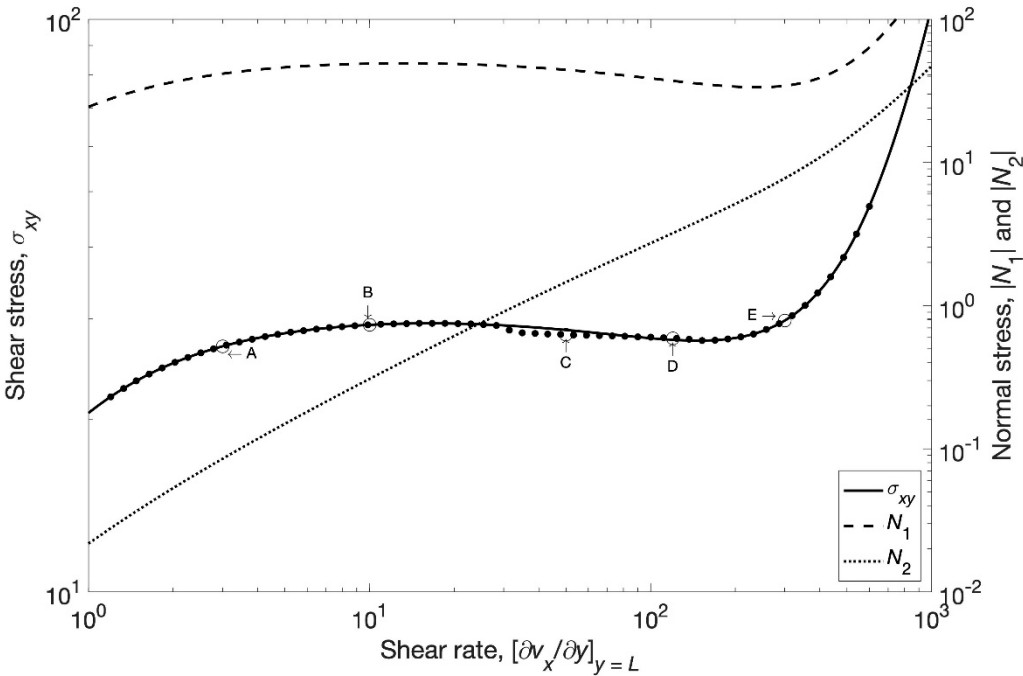

**Figure 1.** Steady state constitutive curves obtained under controlled shear rate starting from rest. Parameters used in the simulations are: $c_0 = 5$ wt %, $\varphi_\infty = 847$ m·s·kg$^{-1}$, $\tau_\sigma = 1.06$ s, $\eta_0 = 36.56$ kg m$^{-1}$s$^{-1}$, $k_0 = 3.28 \times 10^{-4}$ m s$^{-1}$ kg$^{-1}$, $\lambda = 0.136$ s, $\vartheta = 0.0061$ s, $\rho = 1000$ kg m$^{-3}$, $\mathcal{D} = 1 \times 10^{-5}$ m$^2$ s$^{-1}$, $\mathcal{D}' = 1 \times 10^{-11}$ m$^2$ s$^{-1}$, $\psi_2 = 0.0388$ s.

Figure 2a–g depict the dynamics and attainment of steady state under controlled shear rate when the reference shear rate is 3 s$^{-1}$ (see point A in Figure 1). At this shear rate, the flow is homogeneous, and the steady state is reached at the time scale of the Maxwell relaxation time. The position $y = L$ corresponds to the moving plate and $y = 0$ refers to the fixed boundary. Figure 2g depicts uniform concentration throughout the geometry. The shear stress increases monotonically (Figure 2e) in the same form observed in the fluidity, while first and second normal stress differences are depicted in Figure 2b,c,f, respectively, and a velocity profile corresponding to a single shear rate is observed in Figure 2d.

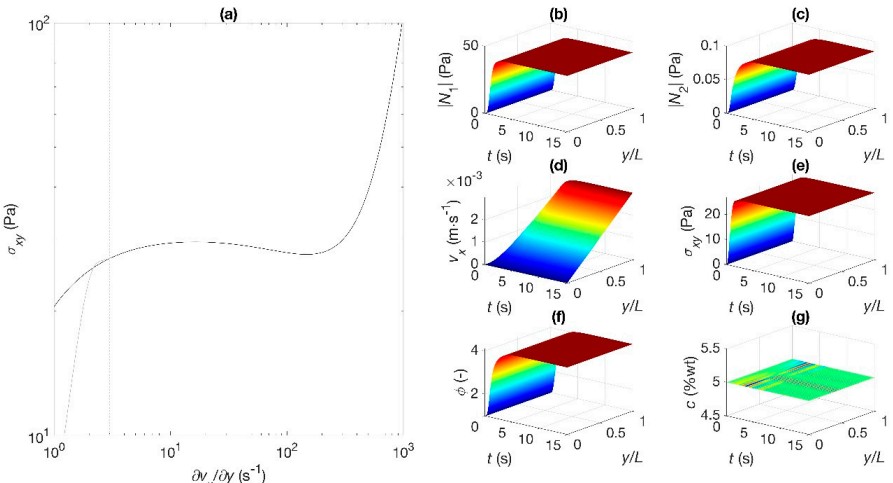

**Figure 2.** Dynamics and steady state under controlled shear rate. Reference shear rate is 3 s$^{-1}$. Temporal trajectory of the stress and shear rate for various spatial positions (**a**), evolution of the normal stress differences $N_1$ (**b**) and $N_1$ (**c**), velocity (**d**), shear stress (**e**), fluidity (**f**), and concentration (**g**), in space and time.

In Figure 3a–g, the reference shear rate has been increased to $10 \text{ s}^{-1}$ (point B in Figure 1) corresponding to the top-jumping stress. The dynamics shown in Figure 3a are different to that in the homogeneous region, although the attainment to steady state is fast. The trajectory before steady-state includes an overshoot before landing at the top jumping stress. Once again, the concentration is uniform (Figure 2g) and the fluidity and second normal stress difference grow monotonically (Figure 2c,f, respectively). In contrast, the shear stress (Figure 2e) and the first normal stress difference (Figure 2b) present an overshoot at the inception of flow before they reach steady state. The velocity profile corresponds to a single shear rate (Figure 2d).

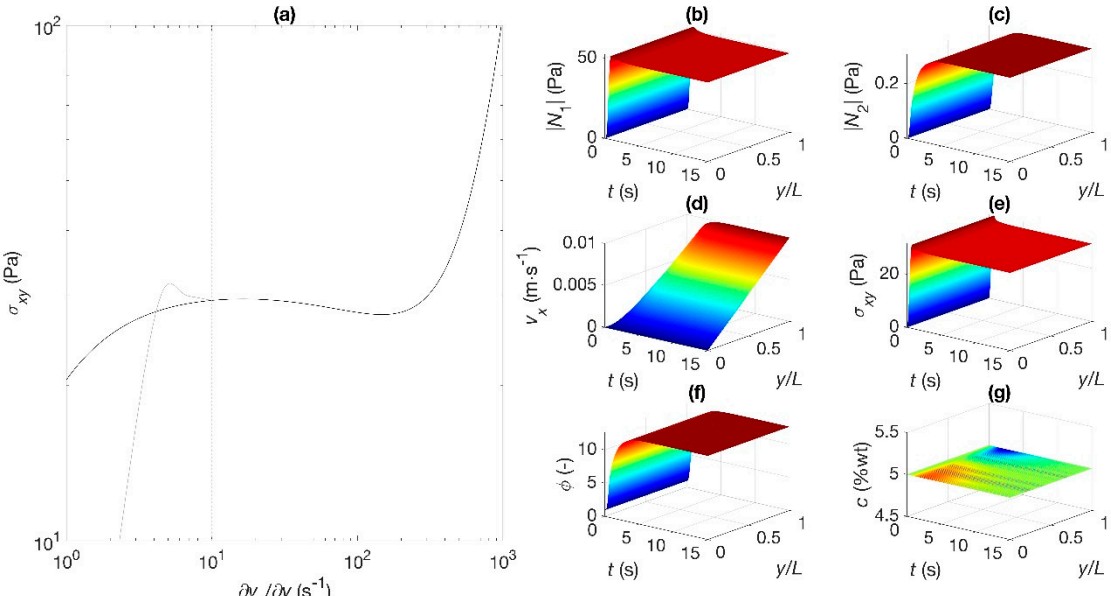

**Figure 3.** Dynamics and steady state under controlled shear rate. Reference shear rate is $10 \text{ s}^{-1}$. Temporal trajectory of the stress and shear rate for various spatial positions (**a**), evolution of the normal stress differences $N_1$ (**b**) and $N_2$ (**c**), velocity (**d**), shear stress (**e**), fluidity (**f**), and concentration (**g**), in space and time.

A different situation is shown when the reference shear rate increases to $50 \text{ s}^{-1}$, within the shear banding region (point C). The dynamics now oscillate between the two attractors located at the critical shear rates (at the binodals or extremes of the plateau stress) after describing an overshoot in the stress (Figure 4a). The concentration is not uniform (Figure 4g) and transient and steady state banding is predicted as the velocity profile changes as a function of time from a single into a two-banded profile corresponding to two shear rates, with the steepest one located next to the moving plate (Figure 4d), within the region where the concentration decreases. A depletion zone then appears near the moving plate in the high shear rate region resulting in a concentration gradient; the fluidity in turn increases next to the moving plate (Figure 4f). As before, the shear stress (Figure 4e) and $N_1$ (Figure 4b) describe an overshoot at the inception of flow, but rapidly they attain steady state. A remarkable result in both stress differences is the decrease in $N_1$ simultaneous to an increase in $N_2$ in the region next to the moving plate (Figure 4b,c, respectively). In fact, a jump in both stress differences across the interfaces is revealed in these predictions.

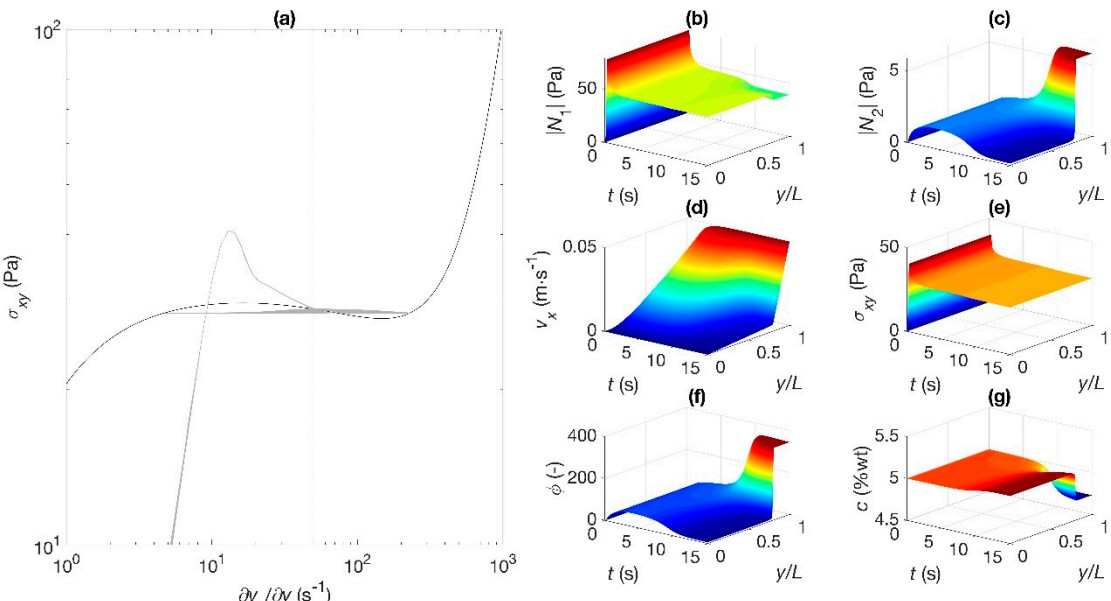

**Figure 4.** Dynamics and steady state under controlled shear rate. Reference shear rate is 50 s$^{-1}$. Temporal trajectory of the stress and shear rate for various spatial positions (**a**), evolution of the normal stress differences $N_1$ (**b**) and $N_2$ (**c**), velocity (**d**), shear stress (**e**), fluidity (**f**), and concentration (**g**), in space and time.

In Figure 5, the reference shear rate is now 120 s$^{-1}$, which is in the region near the minimum of the flow curve (high shear rate attractor, point D in Figure 1). The dynamics rapidly converge to the extreme of the plateau stress (Figure 5a) after an overshoot and oscillations along the plateau. Once again, concentration gradients are predicted, including a sudden decrease of concentration at the interface (Figure 5g) inducing a sudden rise in the fluidity (Figure 5f) next to the moving wall. The high shear rate band covers most of the flow region (Figure 5d) and past a pronounced maximum the total stress is uniform along the flow cell (Figure 5e). As found in the shear banding region, there is a jump in the normal stresses across the interface (Figure 5b,c), but $N_1$ develops this sudden change after a short overshoot, in contrast to $N_2$, which monotonically increases in the high shear rate band and decreases to almost zero in the low shear rate band.

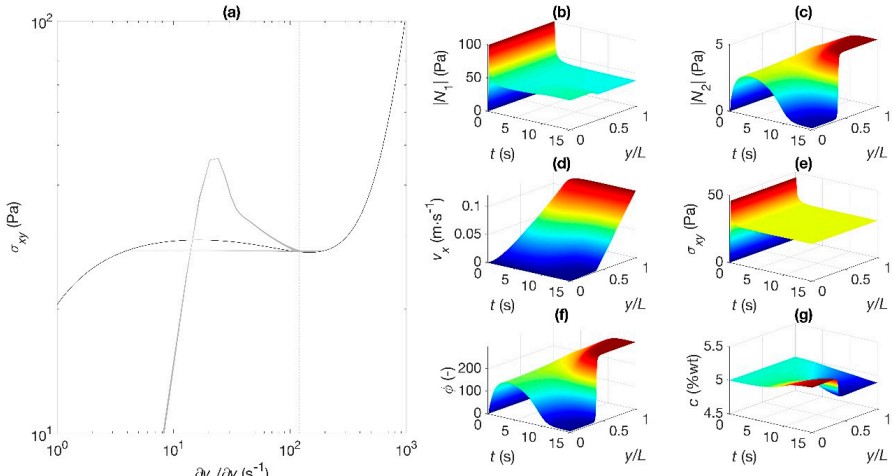

**Figure 5.** Dynamics and steady state under controlled shear rate. Reference shear rate is 120 s$^{-1}$. Temporal trajectory of the stress and shear rate for various spatial positions (**a**), evolution of the normal stress differences $N_1$ (**b**) and $N_2$ (**c**), velocity (**d**), shear stress (**e**), fluidity (**f**), and concentration (**g**), in space and time.

Finally, in Figure 6a–g, the reference shear rate is 300 s$^{-1}$ located in the high shear-rate branch (point E in Figure 1). Past an overshoot, the transient dynamics ends at the high shear-rate branch of the flow curve. The flow is again homogeneous, since a single velocity gradient is predicted (Figure 6d). Concentration gradients are absent at steady state and fluidity is uniform again (Figure 6f,g). The shear stress and $N_1$ rapidly attain steady state after maxima following the inception of flow (Figure 6b); $N_2$ attains steady-state monotonically (Figure 6c).

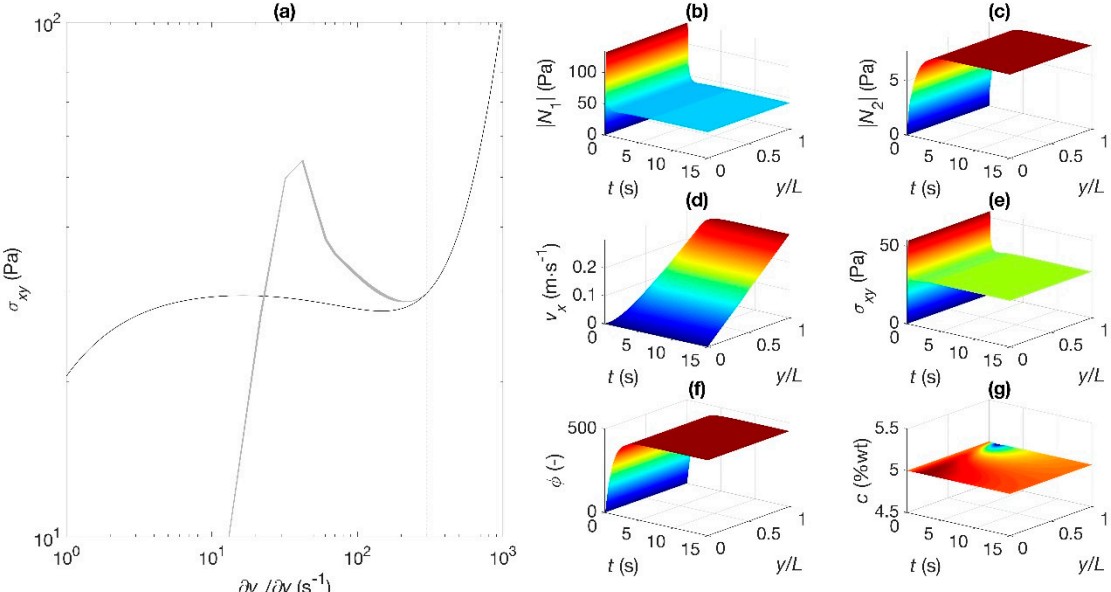

**Figure 6.** Dynamics and steady state under controlled shear rate. Reference shear rate is 300 s$^{-1}$. Temporal trajectory of the stress and shear rate for various spatial positions (**a**), evolution of the normal stress differences $N_1$ (**b**) and $N_2$ (**c**), velocity (**d**), shear stress (**e**), fluidity (**f**), and concentration (**g**), in space and time.

## 4. Discussion and Conclusions

The model presented here contains three constitutive equations: the equation for the stress, for the structural parameter, and for the diffusion of mass. They are mutually-coupled by phenomenological coefficients, and their physical significance arises as they identify with the mechanisms acting on the system. The mechanisms included here are the kinetics involved in the attainment of steady state, after which a banding state is induced by the non-monotonic nature of the constitutive equation. The existence of a banded state induces jumps in the normal stresses and concentration differences amidst the bands, which constitute important predictions of the model. The diffusion of structure and mass are governed by the diffusion coefficients, as it should be in situations where concentration gradients or "structure gradients" arise. The latter is a consequence of the existence of ordered phases or bands coexisting with more disordered or isotropic bands in the flow cell.

Instabilities leading to undulations of the interface [9,10] appear from the action of normal stresses across the interface between the bands. Accordingly, the mechanism of instability is not fully understood, but it is likely to stem from steep gradients in the normal stress and shear rate across the interface. Various authors have found instabilities due to the jump of normal stresses across the interface in polymeric systems [30,31]. Alternating vorticity bands [32] and concentration gradients in polymer solutions that exhibit shear banding have been predicted.

One of the advantages of the present model is that it describes the breakage-reformation process of the micelles under flow so as to enable an understanding of the relation between shear band formation

and microstructure evolution. In the BMP model, the relation between the fluidity, relaxation time $\tau$, and micellar length $n$ is [24]:

$$\tau = \frac{1}{G_0 \varphi} = \tau_0 \left( \frac{n}{n_0} \right)$$

where $\tau_0$ is the relaxation time when $\varphi = \varphi_0$. Substitution of this equation into the evolution equation for the fluidity (see Equation (1)) yields an equation expressed in terms of the average micellar length, which is the microstructural variable. The physical interpretation is that the micellar length n follows an evolution equation related to the breakage and reformation process of the micelles.

Results of the present model and those contained in reference [24] (see Figure 5 therein), where no normal stresses are included, reveal predictions of multiple bands in the absence of normal stresses. In fact, a comparison at similar imposed shear rates (those near the high shear-rate extreme of the plateau) of the velocity and stress fields are similar, but the fluidity presents two regions with high fluidity next to the walls (without normal stresses). With normal stresses, we observe a single region of high fluidity next to the moving wall (see Figure 5 of the present paper).

We have shown that the EIT formalism described here is consistent with previous works on shear-banding inhomogeneous flows. In fact, in the two-fluid approach, the viscoelastic stress causes the micelles to diffuse up in gradients of the stress, and so it couples stress with concentration. In the present work, predictions of a depleted layer next to the moving surface reveal agreement with this underlying mechanism, i.e., the existence of a positive feedback, which causes diffusion of micelles up their own concentration gradient. As an explanation to micellar migration [21], the strain component $W_{yy}$ is more negative in the high shear rate phase than in the lower shear phase, then micelles migrate to the low-shear band. This corresponds to the strongly sheared micelles stretched strongly along the flow direction.

A model for wormlike micellar solutions involving scission and reforming of chains based on non-affine network theory and a discrete version of the Cates theory was forwarded [16]. Although the model does not predict $N_2$, one of the variants of the model (PEC + M, partial extended convected derivative with two interacting species) predicts a behavior of $N_1$ as a function of shear rate quantitatively similar to that predicted by the present model. In reference [16], the first normal stress difference also shows a banded structure and a sudden drop at the interface. Predictions of the overshoot at the onset for flow follow the same manner as predictions by the BMP model.

Further concordance with predictions of other models arises. The stress overshoot predicted in the banded state in Figure 4 agrees with the three stages predicted by the non-local JS model after a step growth in shear rate [33], i.e., band destabilization, interface reconstruction, and interface traveling. As indicated in this reference, the instability and reconstruction of the interface in the first two stages end when the interface between stable bands sharpens. Front propagation is controlled by the diffusion constant D, as in the BMP model.

Predictions of the concentration profiles in entangled polymeric systems [34] depict a sudden decrease (quasi-step like) of concentration at a given position in the flow cell, and this change occurs nearer to the moving wall when the shear rate is smaller. These predictions agree qualitatively with those depicted in Figures 4 and 5 of the present paper. Band migration and band shapes are similar in both systems, illustrating that this phenomenon is common to wormlike micelles and entangled polymeric systems

In summary, the relevant predictions of the present model are the depleted concentration region near the moving boundary and the jumps in normal stresses across the interface. Stress diffusion arises naturally in the constitutive equations. Two diffusion mechanisms are involved, the mass and the structural diffusion, which arise in the equations for the stress and stress differences, but in addition, they are present in the equation for the reformation-breakage of the structure.

**Author Contributions:** All four authors (J.P.G.-S., F.B., J.E.P., O.M.) participated in the theoretical descriptions. J.E.P. and O.M. undertook the editing process.

**Funding:** This research was funded by CONACYT (National Council for Science and Technology, Mexico).

**Conflicts of Interest:** The authors declare no conflict of interest. Sponsors had no role in this study, nor in the writing of the manuscript, or decision to publish the results.

## Appendix A

Extended Thermodynamic description of complex fluids (see [4]). The non-equilibrium thermodynamic state of a complex fluid is described by a formulation contained in the usual procedure of Extended Irreversible Thermodynamics (EIT). The thermodynamic state shall be described by taking as conserved variables {C} the internal energy density (e), the mass density ($\varrho$), and the relative concentration of the dispersed phase (*c*) that is embedded in a Newtonian liquid of concentration $c'$, i.e., $c + c' = 1$. As for the set {R} of non-conserved state variables, a scalar representing the internal structure of the fluid $\varsigma$, the diffusive concentration flux **J**, and the traceless symmetric part of the stress tensor **σ** are included. Hence, the space of state variables for this system is given by the set G = C U R = {e, $\varsigma$, *c*; $\rho$, $\bar{J}$, $\underline{\sigma}$}. As the first basic assumption of the theory, EIT assumes the existence of a sufficiently continuous and differentiable function $\eta_E$, defined over a complete space G: $\eta_E = \eta_E$ {e, $\varsigma$, *c*; $\rho$, $\bar{J}$, $\underline{\sigma}$}. This assumption aims to generate a differential form, which, in a strictly formal sense, will generalize the Gibbs relation of local equilibrium thermodynamics. For an incompressible fluid at constant temperature, applying the usual procedure of EIT to the given generalized-entropy function and restricting the scheme to the lowest order in the non-conserved variables, we obtain the following generalized Gibbs relation:

$$T\frac{\mathrm{d}\eta_E}{\mathrm{d}t} = -\mu\frac{\mathrm{d}c}{\mathrm{d}t} + \frac{1}{\rho}\alpha_0\frac{\mathrm{d}\varsigma}{\mathrm{d}t} + \frac{1}{]\rho}\bar{\alpha}_1\cdot\frac{\mathrm{d}\bar{J}}{\mathrm{d}t} + \frac{1}{\rho}\underline{\alpha}_2 : \frac{\mathrm{d}\underline{\sigma}}{\mathrm{d}t} \tag{A1}$$

here *T* and $\mu$ are the local equilibrium values of the temperature and the chemical potential of the dispersed phase; $\alpha_0$, $\bar{\alpha}_1$, and $\underline{\alpha}_2$ are phenomenological coefficients that are defined as the partial derivatives of $\eta_E$ with respect to the state variables, and hence, depend on the equilibrium value of *c*. The scalar $\alpha_0$, the vector $\bar{\alpha}_1$, and the tensor $\underline{\alpha}_2$ should be constructed as the most general scalar, vector, and tensor expressions that may be obtained in terms of all independent variables in G. Thus, according to the theory of invariants in space G and to the first order in the non-conserved variables, they are given by:

$$\underline{\alpha}_0 = \alpha_{00}\zeta, \quad \underline{\alpha}_1 = \alpha_{10}\underline{J}, \quad \underline{\alpha}_2 = \alpha_{20}\underline{\sigma} \tag{A2}$$

where $\alpha_{i0}$ (*i* = 0, 1, 2) are scalar coefficients. It should be stressed that these phenomenological coefficients can only be determined from experiment or from a microscopic theory.

The second postulate of EIT assumes that the function $\eta_E$ satisfies a balance equation, namely,

$$\rho\frac{\mathrm{d}\eta_E}{\mathrm{d}t} + \nabla\cdot\bar{J}_\eta = S_\eta \tag{A3}$$

$\bar{J}_\eta$ and $S_\eta$ denote the flux and source term associated to $\eta_E$, respectively. They should be expressed as the most general vector and scalar in G. Consistency with the order considered in arriving at Equation (A1) requires that

$$\bar{J}_\eta = \beta_0\varsigma\bar{J} + \beta_1\bar{J} + \beta_2\bar{J}\cdot\underline{\sigma} \tag{A4}$$

$$S_\eta = X_0\varsigma + \overline{X}_1\cdot\bar{J} + \underline{X}_2 : \underline{\sigma} \tag{A5}$$

where the phenomenological coefficients $\beta_i$ depend on the local equilibrium value of *c*. Furthermore, we consider $X_i$, up to first order, as the most general quantities in G. It is important to point out that in order to recover the usual results of the linear irreversible thermodynamics (LIT) near equilibrium, $\eta_E$, $\bar{J}_\eta$, and $S_\eta$ should reduce to the entropy production, the entropy flux, and the entropy production, respectively. Therefore, $\beta_1 = -\mu/T$.

By computing the divergence of Equation (A4) and using Equation (A1) and the mass conservation equation, we get:

$$\rho \frac{dc}{dt} = -\nabla \cdot \overline{J} \tag{A6}$$

The following explicit expression is obtained from Equation (A3):

$$\rho \frac{d\eta_E}{dt} + \nabla \cdot \overline{J}_\eta = \frac{\varsigma}{T}\left[\alpha_{00}\frac{dc}{dt} + \beta_0 T \nabla \cdot \overline{J}\right] + \frac{1}{T}\overline{J} \cdot \left[\alpha_{10}\frac{d\overline{J}}{dt} - T\nabla(\frac{\mu}{T}) + \beta_0 T\nabla\varsigma + \beta_2 T\nabla \cdot \underline{\underline{\sigma}}\right] \\ + \frac{1}{T}\underline{\underline{\sigma}} : \left[\alpha_{20}\frac{d\sigma}{dt} + \beta_2 T\nabla\overline{J}\right] \tag{A7}$$

By substituting Equation (A5) into Equation (A3) and using Equation (A7), the following coupled relaxation equations for the non-conserved variables are obtained:

$$\tau_0 \frac{d\varsigma}{dt} = -X_0 + \beta_0 \nabla \cdot \overline{J} \tag{A8}$$

$$\tau_1 \frac{d\overline{J}}{dt} = -\overline{X}_1 - \nabla\mu + \beta_0 \nabla\varsigma + \beta_2 \nabla \cdot \underline{\underline{\sigma}} \tag{A9}$$

$$\tau_2 \frac{d\underline{\underline{\sigma}}}{dt} = -\underline{\underline{X}}_2 + \beta_2 \nabla\overline{J} \tag{A10}$$

It is important to stress that in Equation (A5), $S_\eta$ may also depend on the parameters that lie outside G, but which are essential to specify the non-equilibrium state of the system. For the model under consideration, the velocity gradient tensor $\nabla \overline{v}$ is required to formulate the constitutive equations for the stress tensor. Therefore, the expressions for $X_0$, $\overline{X}_1$, and $\underline{\underline{X}}_2$ are written in terms of the non-conserved variables and $\nabla \overline{v}$. With these considerations, the explicit form of Equations (A8)–(A10) is given in Equations (1)–(3).

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
