# Peer review of "Inhomogeneous Flow of Wormlike Micelles: Predictions of the Generalized BMP Model with Normal Stresses"

_fluids, doi:10.3390/fluids4010045_

Round 1

Reviewer 1 Report

This is an excellent paper in formulation, execution, analysis and presentation. The topic is of great current interest to the complex fluids community. The topic and results fit very well in this journal.  The material system under study has potential applications in the energy, materials processing, and biomedical fields. The results connecting flow behaviour struture formation and normal stress is novel and significant. I recommend publication. I do not need to revise again.

Very minor points.

There is restrictions on the parameters appearing in the dissipation function.

Given possible different time reversal properties, there may be non-dissipative reactive parameters.

Under steady shear, could a plot of normal stress as a function of shear stress plateau to a value that reveals material parameters.

The vortex-interface instability mentioned in the paper does it relate to Batchelor vorticity-curvature relation?

Author Response

We thank the reviewer for the encouraging comments on the manuscript. The minor point answers follow:

1.- Indeed, there are restrictions to the dissipation function. As the fluidity tends to the limit at high strain rates, the dissipation becomes unbounded, unless a Newtonian limiting fluidity appears in the formulation. This is especially important in extensional flows.

2.- Non-dissipative reactive parameters are analyzed in a recent publication (see ref. AICHEJ 64 (2018) 2277-2292).

3.- We agree with the comment concerning the plot of shear and normal stresses in the plateau region. It may reveal material parameters dealing with the time scale formation and span of the plateau.

4.- The so-called vorticity bands are here referred to the appearance of alternating bands occurring along the axis of a Couette flow cell, normal to the shear plane, rather than Batchelor vorticity curvature relation.  

Author Response

1.- We have explored different expressions for the destruction function, depending upon the system nature, namely, shear-thinning or shear-thickening, thixotropic, rheopectic, and so on. In general, ko may be a function of the invariants of the strain rate or stress.

The equation postulates a linear relation of the relaxation time (which is itself a function of the fluidity) with the micellar length. Therefore, the fluidity and the micellar length are inversely proportional, and the equation for the micellar length is similar to that of the shear viscosity.

2.-  Re: Indeed, as stated in the manuscript and in Ref. [4], all of the model parameters are obtained with simple experiments described in Ref. [4]. The values of these parameters are surfactant type and concentration dependent.  However, once they are available, the same values are valid in other flow situation with acceptable to excellent accuracy in the BMP model predictions.   

Reviewer 3 Report

This paper is a numerical study of the generalized BMP model, in particular, with shear banding and normal stresses. The BMP model is presented in a similar fashion as the various prior works on the model. A lot of work has been done over the years with the BMP model, from simple shear to complex flow calculations. The main results presented here are the predictions of normal stress and concentration gradients produced by the model in shear-banded flow. However, the prediction of such gradients is not unique to this work, in fact there has been a lot of work on this over the last 20 years with various model formulations (there are a lot of relevant references absent here). The results presented in this work are limited in scope, with only one set of parameters considered and only 5 flow conditions. No comparisons are made between the current predictions and those of other models, nor a comparison in the BMP model with and without normal stresses.  Furthermore, there is a lack of references to the works in the area of modeling wormlike micelles.

Overall, the paper lacks in its originality, novelty and significance, simply because the results they present have already been reported for many other models. This can be improved by comparing and contrasting with other models, and discussing why the new predictions are important relative to all of these works.

Below I present a list of items for the authors to address that would make this work a more significant contribution:

Top of page 3, it is stated “Comparison of the free energy calculated with and without normal stresses is shown elsewhere [25].” However, reference 25 is the paper by Lu, Olmsted and Ball and does not address normal stresses. Since the major contribution of this work is to look at the normal stresses this seems to be an important place to elaborate on the comparison of the model with and without normal stresses.

There has been a lot more recent modeling of wormlike micelles, for example from Antony Beris’ and Mike Graham’s groups, that give different mechanisms for breaking/reforming. See (5)-(9) below. These models present the mechanisms for the breakage of chains, what is the actual mechanism in the BMP model?

There has been a lot more work on concentration coupling in polymeric systems in nonlinear flows, see for example the work from Gary Leal’s and Natalie Germann’s groups, see references (10)-(14) below. These works use more advanced models, and give detailed descriptions of the stress-concentration coupling in shear banding. How does the current model and its predictions compare and contrast with the predictions (other than they are for polymers and not micelles)?

Why present the equations in two-dimensions? Couldn’t the one-dimensional equations be derived from the tensorial form of the equations? This seems rather superfluous.

In Figure 1, what is the dotted curve?

The results presented show a positive value of the second normal stress difference, N2. However, experiments with wormlike micelles (ref 17) show that N2 is negative. This is also true for most polymeric systems (see for example the book The theory of polymer dynamics by Doi and Edwards). Are you plotting the absolute value of N2, or is it indeed positive? If the latter, this would run counter to the explanation that concentration moves up stress gradients.

It is claimed that equation 25 is equal to that derived from equation 26.  Equation 26 involves the Gordon-Showalter derivative, which has an additional parameter, the slip parameter, which the authors do not specify. Furthermore, equation 26 is just the Jonhson-Segalman model with diffusion which was originally proposed by Peter Olmsted (refs (3) and (4) below).

Diffusive motion of the interface has also been seen and discussed in shear banding in other works with other models, for example the JS and VCM  models (refs (1) and (2) below – ref (1) is briefly mentioned in the manuscript).

The very last paragraph in the conclusion discusses the main results of the work: concentration gradient, normal stress jump, natural inclusion of stress diffusion, mass and structural diffusion, and breakage-reformation through the structure. Concentration gradients have been studied in depth in various recent works – how is the current model/results an improvement upon these? As mentioned, normal stress jumps are present in other shear banding models (e.g., Johnson-Segalman, Giesekus, Rolie-Poly, VCM, etc.) – what’s different with the BMP model? Stress diffusion naturally arises in various models, e.g., the VCM model, the model by Germann, Cook and Beris, the original work of El-Kareh and Leal, and in more recent two-fluid models – what are the advantages of the BMP model?

On page 6, it is stated “In the absence of normal stresses, a situation commonly encountered in semi-dilute micellar systems …”. No references are provided to support this statement. In fact, it does not appear to be a true statement (see, for example, (15) below).

There’s a discussion of the role normal stresses play in the instability of shear-banded flows, but no stability calculations are provided here.

It is said on page 9 that the fluidity lowers next to the moving wall for shear rate of 50, but the opposite is shown in figure 4.

What is the value of the nondimensional diffusion coefficient, in order that comparisons can be made with the various other models that included diffusion?

Does the BMP predict hysteresis? What if step down simulations are run?

References:

(1)  Radulescu, O., Olmsted, P.D., Decruppe, J.P., Lerouge, S., Berret, J.F. and Porte, G., 2003. Time scales in shear banding of wormlike micelles. EPL (Europhysics Letters)62(2), p.230.

(2)  Cromer, M., Cook, L.P. and McKinley, G.H., 2011. Pressure-driven flow of wormlike micellar solutions in rectilinear microchannels. Journal of Non-Newtonian Fluid Mechanics166(3-4), pp.180-193.

(3)  Lu, C.Y.D., Olmsted, P.D. and Ball, R.C., 2000. Effects of nonlocal stress on the determination of shear banding flow. Physical Review Letters84(4), p.642.

(4)   Olmsted, P.D., Radulescu, O. and Lu, C.Y., 2000. Johnson–Segalman model with a diffusion term in cylindrical Couette flow. Journal of Rheology44(2), pp.257-275.

(5)  Dutta, S. and Graham, M.D., 2018. Mechanistic constitutive model for wormlike micelle solutions with flow-induced structure formation. Journal of Non-Newtonian Fluid Mechanics251, pp.97-106.

(6)  Germann, N., Kate Gurnon, A., Zhou, L., Pamela Cook, L., Beris, A.N. and Wagner, N.J., 2016. Validation of constitutive modeling of shear banding, threadlike wormlike micellar fluids. Journal of Rheology60(5), pp.983-999.

(7)  Germann, N., Cook, L.P. and Beris, A.N., 2016. A differential velocities-based study of diffusion effects in shear banding micellar solutions. Journal of Non-Newtonian Fluid Mechanics232, pp.43-54.

(8)   Germann, N., Cook, L.P. and Beris, A.N., 2014. Investigation of the inhomogeneous shear flow of a wormlike micellar solution using a thermodynamically consistent model. Journal of Non-Newtonian Fluid Mechanics207, pp.21-31.

(9)   Germann, N., Cook, L.P. and Beris, A.N., 2013. Nonequilibrium thermodynamic modeling of the structure and rheology of concentrated wormlike micellar solutions. Journal of Non-Newtonian Fluid Mechanics196, pp.51-57.

(10) Cromer, M., Villet, M.C., Fredrickson, G.H. and Leal, L.G., 2013. Shear banding in polymer solutions. Physics of Fluids25(5), p.051703.

(11) Cromer, M., Fredrickson, G.H. and Leal, L.G., 2014. A study of shear banding in polymer solutions. Physics of Fluids26(6), p.063101.

(12) Peterson, J.D., Cromer, M., Fredrickson, G.H. and Gary Leal, L., 2016. Shear banding predictions for the two-fluid Rolie-Poly model. Journal of Rheology60(5), pp.927-951.

(13)  Hooshyar, S. and Germann, N., 2016. A thermodynamic study of shear banding in polymer solutions. Physics of Fluids28(6), p.063104.

(14) Hooshyar, S. and Germann, N., 2017. Shear banding of semidilute polymer solutions in pressure-driven channel flow. Journal of Non-Newtonian Fluid Mechanics242, pp.1-10.

(15) Shikata, T., Dahman, S.J. and Pearson, D.S., 1994. Rheo-optical behavior of wormlike micelles. Langmuir10(10), pp.3470-3476.)

Author Response

We thank the reviewer for the insight comments to improve the manuscript. Comparisons suggested using the BMP model with or without normal stresses, and comparisons with other models have been included in the discussion. In addition, although the results exposed here appear in other models, the derivation from extended irreversible thermodynamics and the framework used are original. It is useful to present alternative derivation of the models; the agreement found with predictions from other models, albeit derived differently, justifies the effort.

1.- We have omitted the paragraph related with reference 25. Comparisons suggested between the BMP model with and without normal stresses are included. Results of the present model and those contained in reference 24 (see Figure 5 therein), where no normal stresses are included, reveal predictions of multiple bands in the absence of normal stresses. In fact, a comparison at similar imposed shear rates (those near the high shear-rate extreme of the plateau) the velocity and stress fields are similar, but the fluidity presents two regions with high fluidity next to the walls (without normal stresses). With normal stresses, we observe a single region of high fluidity next to the moving wall (see Figure 5 of the present paper). 

2.- The mechanism of structural breakage in the BMP model involves a kinetic process which is proportional to the total dissipation of the system. More elaborated models that we have worked out (see ref. AICHEJ 64 (2018) 2277-2292), postulate that the kinetics of structure breaking depends on the reverse kinetic constants, which are themselves functions of the thermodynamics of irreversible processes in the system.   

3.- We do not intend to perform an insight analysis on the flow-concentration coupling. Rather, we want to emphasize that a result of the derivation of the constitutive equations is the coupling of the diffusion current with flow, and from this coupling, the concentration gradients arise. As a comparison, in Peterson et al. (Jour. Rheology 60, 5, 927, 2016), predictions of the concentration profiles in entangled polymeric systems depict a sudden decrease (quasi-step like) of concentration at a given position in the flow cell, and this change occurs nearer the moving wall the smaller the shear rate is. These predictions agree qualitatively with those depicted in Figures 4 and 5 of the present paper. Band migration and band shapes are similar in both systems, illustrating that this phenomenon is common to wormlike micelles or entangled polymeric systems.

4.- We have plotted the absolute value of the second normal stress difference. In fact, ψ2 is negative. In creeping shear flows, Eq. (26) reduces to Eq. (25). An explanation is included in the new version.

5.- We present the equations in two dimensions because  the analysis refers to the plane of shear. The tensorial form allows the derivation of the one-dimensional cases.

6.- Interface migration is discussed at length in a recent work (Rheol. Acta 56 (2017) 765-778) with the BMP model. Among the advantages of the model, one of them is its simplicity to calculate complex flow histories, for example, thixotropic and rheopectic loops, and others, introducing in a clear form the kinetics of breakage-reformation of the micelles, or alternatively, those of the entanglement-disentanglement process. Moreover, it allows incorporating complex kinetics and complex geometries, within an irreversible thermodynamics framework. Of course, more elaborated models can predict complicated flow histories. One of the extensions of the model considered complex kinetics with a clear connection to microscopic thermodynamic variables (see ref. AICHEJ 64 (2018) 2277-2292). In that work, a quantitative prediction of the first normal stress difference is highlighted, which rarely has been predicted by other models.

7.- The model involves predictions of N2, which is normally neglected in many analyses. The evolution of N2 with space and time and that of the other rheological variables is the result of the coupling of the stress, the diffusion current and the structural modifications induced by flow and diffusion. Although stress diffusion naturally arises in various models, here we demonstrate that not only the stress diffusion is important, but also the structural diffusion and the mass diffusion, with mutual couplings.

8.- The paragraph in Page 6 is corrected. In semi-dilute micellar solutions, normal stresses are small, but have important effects on the stability of the system. We mentioned that normal stresses influence the flow stability and provide with some examples, but with no involvement in this issue. A stability analysis on the BMP model appeared in ref. (JNNFM, 144 (2007) 160-169).

9.- We corrected the statement in Figure 4. In fact, the fluidity increases near the wall. The magnitude of the two diffusion coefficients appears in the legend of Figure 1.

10.- The model can predict hysteresis, with and without normal stresses (see ref. 24). We can explore many flow histories, but with large number of added figures.

Round 2

Reviewer 3 Report

I thank the authors for their work addressing my comments, and I support publication of the paper.  I have a few minor comments:

1) For clarity of presentation, I think N_2 in the figures should appear as |N_2|, to eliminate any confusion.

2) The shear rate gamma_dot, I don't believe, is ever defined in the manuscript.

Author Response

We have defined the shear rate and we clearly define the absolute value of the stress differences.

Thank you for the corrections.